# Obtaining N-Enriched Mesoporous Carbon-Based by Means of Gamma Radiation

**DOI:** 10.3390/nano12183156

**Published:** 2022-09-12

**Authors:** Piotr Kamedulski, Malgorzata Skorupska, Izabela Koter, Maciej Lewandowski, Víctor Karim Abdelkader-Fernández, Jerzy P. Lukaszewicz

**Affiliations:** 1Faculty of Chemistry, Nicolaus Copernicus University, Gagarina 7, 87-100 Torun, Poland; 2Centre for Modern Interdisciplinary Technologies, Nicolaus Copernicus University, Wilenska 4, 87-100 Torun, Poland; 3Department of Inorganic Chemistry, University of Granada, 18071 Granada, Spain

**Keywords:** carbon materials, mesoporous activated carbons, gamma irradiation, heteroatoms, nitrogen

## Abstract

In this paper, we present the results of the gamma irradiation method to obtain N-doped mesoporous activated carbons. Nitrogen-enriched mesoporous carbons were prepared from three chosen commercial activated carbons such as Carbon Black OMCARB C-140, KETJENBLACK EC-600JD and PK 1-3 Norit. HRTEM, SEM, Raman spectra, elemental analysis, XPS studies and widely approved N_2_ adsorption–desorption measurements allowed us to evaluate the effectiveness of N atom insertion and its influence on the BET surface area and the pore structure of modified carbons. The obtained materials have an exceptionally high N content of up to 3.2 wt.%. Additionally, selected N-doped activated carbons were fully characterized to evaluate their applicability as carbon electrode materials with particular emphasis on Oxygen Reduction Reaction (ORR). The proposed method is a relatively facile, efficient and universal option that can be added to the already known methods of introducing heteroatoms to different carbons.

## 1. Introduction

Carbon-based materials are still a cutting-edge research issue regarding even such diversified ones as fibers, nanotubes, nano-onions, nano-horns, graphite, graphene (and its derivatives), glassy carbon, pyrolytic char, activated carbon, etc. [1,2,3,4]. One of the main application branches of carbon-based materials refers to its surface properties. One may cite the most typical as (i) the adsorption of selected chemical species from the gas and liquid phase for environmental and purification purposes, and (ii) electrode manufacturing for electrochemical devices for generating energy (fuel cells) and/or energy storage (capacitors, supercapacitors and primary/rechargeable batteries). Recently, due to the increasing importance of electric energy storage, most attention has been paid to the eletrocatalytic properties (ORR) of N-doped carbon materials working as potential cathodes in fuel cells and air–metal batteries [5,6,7]. The mentioned practical use strongly depends on such factors as surface area velocity, pore volume, pore size distribution, and above all the presence of adsorption/catalytic domains. Some of these key features appear spontaneously during the manufacturing process as the thermal conversion (carbonization) of an organic N-rich precursor (such as chitosan, polyacrylonitrile, etc.) to a char or activated carbon. However, these spontaneous effects are hardly controllable; therefore, practical applications are limited, with no chance for any improvement. Progress is, however, possible provided naturally formed carbon materials are subjected to appropriate additional processing which may help steer/tailor the abovementioned key features, prioritizing the N content. Still, satisfactory structural parameters (such as BET surface area, and well-developed pore structure) should be ensured.

Generally, two main processing categories should always be considered: the creation of heterogeneous adsorption/catalytic centers and/or tailoring the material’s structural parameters such as the specific surface area and pore structure (including pore size distribution). Moreover, carbon materials in any form can be chemically modified by the insertion of different heteroatoms such as nitrogen, oxygen, boron, phosphorus, sulfur or metal/metal oxide clusters. Nitrogen doping is one of the most popular chemical modifications of carbons due to its potential electrochemical application [8,9,10].

Nitrogen enrichment is commonly regarded as the most effective way to eliminate noble metals (mostly platinum and palladium) from electrodes for electrochemical energy generation/storage devices such as photovoltaic cells, supercapacitors, fuel cells, metal–air batteries and Li-on batteries [11,12,13,14,15,16]. Activated carbons, having rather chemically reactive surfaces (spontaneous self-oxidation in air), relatively easily undergo the insertion of heteroatoms (including nitrogen). In contrast, carbon materials predominantly consisting of aromatic carbon six-ring domains (as in graphene and graphite, CNTs, etc.) are chemically stable and hardly undergo the chemical insertion of heteroatoms (Table 1) [17,18,19,20,21,22]. Thus, most of the enrichment methods suitable for activated carbons are not applicable to graphene or graphite Therefore, there is an obvious need to develop more universal methods for nitrogen insertion applicable to possibly all carbon-based materials. As demonstrated in our previous work, the influence of some key parameters, such as the porous structure, carbonization temperature and presence of nitrogen, at the same time does not always have a positive effect on the electrochemical efficiency of the obtained carbons in the ORR [16,17]. In our previous text, we compiled several methods, which are commonly exploited, for example, N-enrichment processes, i.e., nitrogenized of a slightly reactive graphene/graphite matrix [17] (Table 1).

In the current study, the authors intend to verify the gamma radiation method in terms of commercial activated carbons enriched in nitrogen. The authors plan to use commercial mesoporous activated carbon due to practical reasons, i.e., mass production of N-enriched activated carbons for electrochemical applications. The authors do not consider laboratory-scale activated carbons because such materials, despite having interesting properties, are manufactured in small quantities and at an unknown price. These factors exclude such “laboratory”-activated carbons from any large-scale application. Our experiment includes only mesoporous activated carbons, since this type of pore structure is of particular importance due to its liquid electrolytes.

These methods (Table 1) suffer from many technological drawbacks, among which extremely low efficiency appears to be the main obstacle to any scale-up. Another disadvantage is the expected high consumption of hazardous chemicals and chemical waste production as synthesis by-products.

A new method of inserting nitrogen atoms into carbon matrixes is the so-called gamma irradiation method, which was first proposed for graphene nanoplatelets by some authors [17]. The obtained materials had an exceptionally high nitrogen content of up to 4 wt.%. Moreover, such N-doped graphene structures have nitrogen atoms exclusively bonded as quaternary groups. Such carbons are utilized for electrode design in electrochemical devices. Hence, we had the idea to try to study other carbon structures, such as activated carbons, with the irradiation method, the results of which are presented in this paper.

The research hypothesis to be verified is that highly energetic factors, such as gamma radiation, also trigger the formation of highly reactive species on the surface of activated carbons. It is assumed that gamma irradiation of graphitic structures will generate active sites on the surface of carbon matrixes, which are normally chemically inert and stable. In addition, active radicals are created in any media (a liquid or gaseous carrier of heteroatoms) in contact with a different carbon surface and also gamma-irradiated. Thus, active radicals of these two origins (localized on the carbon surface and in the contacting media) may react, yielding new species on carbon surfaces, even on those which are hardly reactive, such as graphene.

Thus, the experiments in gamma radiation-based N-doping proved that the method is effective; however, it is not as spectacular as in the case of other doping methods and materials [33,34]. Those very effective and spectacular methods mostly rely on a complete synthesis of N-rich activated carbons, i.e., the high N-content was achieved by carbonization of N-rich biopolymers (chitin, chitosan, amino acids, etc.), a concept very far from our concept. In the current paper, we aim to modify (N-doping) of already manufactured typical ACs. While it is truth that N-doping may be achieved by a high-temperature gas-phase ammonia treatment [35], ammonia treatment suffers from two basic shortcomings: high mass loss and the emission of toxic hydrogen cyanide (HCN). We are searching for (and testing the limits) an alternative method, i.e., a gamma-radiation-based one.

## 2. Materials and Methods

N-doped activated carbons were synthesized by a self-designed gamma irradiation method [17]. Firstly, 1 g of OMCARB C-140 (two series) purchased from Alchem (Torun, Poland) was added to plastic test tubes and left open. The tert-butylamine purchased from Sigma-Aldrich (Poznan, Poland) was placed in a separate and open test tube. The amine creates some hazard to human health and the environment, but it was selected due to its high chemical reactivity and volatility, which is important since the process is supposed to be a gas-phase. The butylamine hazard may be reduced by the application of ammonia in a water solution, but it would a change the process to a liquid-phase one. All test tubes were sealed in a square transparent plastic box. Then, we inserted the box into a radiation device. Irradiation, as described in our previous work [17], was carried out in a radiation device RChM-γ-20, where the source of gamma radiation is Co-60. The unit has one central irradiation chamber and six side chambers. The main chamber, used for all experiments, was an air-filled cylinder with internal dimensions of 15 cm in diameter and 24 cm in height; 27 Co-60 sources, each 8 cm long and 1.1 cm in diameter (ca. 4 L working volume), were set in an annular arrangement in a ring-shaped cassette around the central chamber. The shielded elevator assembly moved inside the tube to lower the container with samples to the irradiation position for irradiation. Due to the short half-life of Co-60 (2.56 years), the dose rate, determined by a Fricke dosimeter, varied slightly over time during the experiment and ranged from 15.9 to 17.4 Gy/h. Next, each sample was dried in an electric furnace at 50 °C for 2 h. The same procedure was performed for the other used materials: KETJENBLACK EC-600JD, purchased from AkzoNobel (Warsaw, Poland), and Norit carbon (Fluka Analytical) PK 1-3 Norit.

The N-doped samples obtained by the proposed method were denoted as X-Y-I, where, e.g., X-Y—means the type of activated carbon used (OMCARB C-140-AC–O; KETJENBLACK-AC–KB; PK 1-3 Norit-AC–N) I or II—means the dosing rate for the gamma irradiation method (1—means 2.80 kGy; 2—means 5.59 kGy). AC-Y_raw—means raw material before processing.

The morphology of the N-doped activated carbons was analyzed by scanning electron microscopy (SEM, 1430 VP, LEO Electron Microscopy Ltd., Oberkochen, Germany). The obtained carbons were also examined by high-resolution transmission electron microscopy (HRTEM, FEI Europe production, model Tecnai F20 X-Twin, Brno, Czech Republic). The materials obtained prior to the HRTEM microscopic analysis were dispersed in ethanol and treated with an Inter Sonic IS-1K bath for 15 min and deposited on holey carbon-coated copper grids. The volumetric elemental composition (carbon, nitrogen and hydrogen) of the materials was analyzed by means of a combustion elemental analyzer (Vario MACRO CHN, Elementar Analysensysteme GmbH, Langenselbold, Germany). Raman spectra were obtained by a micro-Raman spectrometer (laser wavelength 532 nm, Senterra, Bruker Optik, Billerica, MA, the USA). The laser was tightly focused on the sample surface through a 50× microscope objective. To prevent any damage to the sample, excitation power was fixed at 2 mW. The resolution was 4 cm^−1^, CCD temperature 223 K, laser spot diameter 2.0 µm and total integration time 100 s (50 × 2 s). X-ray photoelectron spectroscopy (XPS, PHI5000 VersaProbe II Scanning XPS Microprobe, Chigasaki, Japan) measurements were performed using a monochromatic Al Kα X-ray source. Survey spectra were recorded for all samples in the energy range of 0 to 1300 eV with a 0.5 eV step; high-resolution spectra were recorded with a 0.1 eV step. The porous structure of the N-doped activated carbons was analyzed at −196 °C using an automatic adsorption instrument, ASAP 2020 Plus (Micromeritics, Norcross, GA, USA). Before analysis, carbons were outgassed in a vacuum at 150 °C for 24 h. The surface area was calculated by the Brunauer–Emmett–Teller (BET) model. Moreover, pore size distributions were determined from the nonlocalized density functional theory (NLDFT) method using the SAIEUS program (Micromeritics, Norcross, GA, USA). Nitrogen was selected as an adsorption gas. There is no guaranteed method for pore size distribution determination for activated carbons (generally: for porous solids), and some studies suggest the use of Ar and/or CO_2_ instead of N_2_ in gas porosimetry, pointing out some disadvantages of the latter gas, especially in the case of ultramicroporous carbons. However, commercial activated carbons selected for this study do not belong to this category, and the application of N_2_ does not face the abovementioned obstacles. Moreover, basic standardization papers authorized by NIST and IUPAC consider the low-temperature N_2_ adsorption as a reliable method for BET surface area and porosity studies [36,37].

An essential stage of the research is to carry out electrochemical tests in order to determine the potential application of the obtained materials. The electrochemical activity of the produced carbon materials was determined by means of a rotating disc electrode (RDE). The tests were carried out on the Autolab electrochemical analyzer (PGSTAT128N, Ultrecht, The Netherlands). Before starting the tests, the surface of glassy carbon (3 mm, GC) was polished, and then the appropriate amount of the catalyst with a packing of 0.4 mg cm^−2^ was applied, prepared in the following way. A total of 2.5 mg of the resulting catalyst was dispersed in a mixture of distilled water, ethanol and Nafion (0.5 wt.%) for 1 h. The activity of the obtained carbon materials was checked on the basis of the measurements of linear voltammetry (LSV) and cyclic voltammetry (CV) in a three-electrode system. The glassy carbon electrode and the applied catalyst constituted the working electrode; the reference electrode was Ag/AgCl in 3 mol L^−1^ KCl, while the platinum wire served as the counter electrode. All tests were performed in a 0.1 M KOH aqueous solution. The reference material was commercial carbon with a 20 wt.% addition of platinum (20 wt.% of Pt). CV and LSV tests were performed in electrolyte saturated with oxygen and nitrogen to determine the actual activity of the catalyst. CV tests were performed at a scanning speed of 10 mV s^−1^, while the LSV was measured at 5 mVs^−1^,and a rotation speed in the range of 800–2800 rpm. The number of electrons (n) involved in the oxygen reduction reaction was calculated from Koutecky–Levich (K–L) equations by converting the system potential with respect to the hydrogen electrode (RHE).
J^−1^ = J_L_^−1^ + J_K_^−1^ = (Bω^1/2^)^−1^ + J_K_^−1^
(1)
B = 0.62nFC_0_(D_0_)^2/3^ν^−1/6^(2)

The formula uses parameters such as: J or measured current density, JL or current density limiting diffusion and JK is by definition the kinetic current density. The remaining parameters are defined as ω,the angular velocity of the electrode, and n, the number of transferred electrons directly participating in the oxygen reduction reaction. F is a Faraday constant of 96485 C mol^−1^, C_0_ isthe concentration of dissolved oxygen which is 1.2 × 10^−6^ mol L^−1^, the diffusion coefficient of dissolved oxygen is D_0_ (1.9 × 10^−5^ cm^2^ s^−1^), the kinetic viscosity of the electrolyte 0.1 mol L^−1^ KOH is denoted as ν, and it amounts to 0.01 cm^2^ s^−1^. The equations were used to determine the slope of the K–L curve and thus to estimate the number of electrons participating in the ORR.

## 3. Results and Discussion

Selected activated carbons were modified with nitrogen using the gamma irradiation method and fully characterized to evaluate their applicability as carbon electrode materials. Characterization included key features such as elemental composition, surface area and pore structure, morphology and the collection of a standard of electrochemical profiles.

### 3.1. Morphology and Structural Characterization of Carbon Materials

Elemental composition results of all investigated materials are presented in Table 2. The nitrogen content for the obtained samples was in the range of 0.9 wt.% to 3.2 wt.% There was an increase in N-content in each modified sample in relation to the starting material and with an irradiation dose/irradiation time. The C-content in activated carbons was high and in the range of 95.2 wt.% to 96.3 wt.% for AC–O samples, 94.2 wt.% to 97.3 wt.% for AC–KB samples and 85.2 wt.% to 87.0 wt.% for AC–N samples. Moreover, the activated carbons modified with N atoms decreased the carbon content wt.% The difference to 100 wt.%, called a residue, is attributed to a removable element such as oxygen. Raw materials were chemically pure and contained only minor amounts of durable impurities.

The HRTEM technique can give helpful knowledge on the structure of the obtained N-doped activated carbons. Figure 1 presents the HRTEM images of representative samples for all series. The images show the typical structure of activated carbons (an amorphous structure and foam-like character of all samples). As visually observed, the structure of obtained N-rich carbons becomes more ordered. The layers presented in Figure 1 are stacked with smaller apparent interlayer spacing. The raw carbon materials are presented in Appendix A.

All gamma irradiation treated samples used in the synthesis have different structure parameters. Generally, the surface area of the samples changed slightly with the increase of the introduction of nitrogen. The samples from the AC–KB series show the highest surface area in the range of 1556–1837 m^2^/g. In turn, the samples from the AC–O series show the lowest surface area in the range of 223–291 m^2^/g. AC–O and especially AC–KB samples are mesoporous carbons. On the other hand, AC–N samples are micro/mesoporous carbons with a relatively high micropore volume V_mi_ in the range of 0.153–0.256 cm^3^/g.

Figure 2 shows SEM/EDX mapping images of representative samples for all investigated series of activated carbons. The surface and pore entrances are covered pointwise by a tert-butylamine. Nitrogen is present on the entire surface of the carbons. This fact confirms that the proposed irradiation method does not significantly damage the porous structure of the starting material, regardless of the type of activated carbon used. As in the case of our previous obtained N-rich graphene [17], the proposed method is also highly effective in producing N-doped activated carbons.

According to the IUPAC classification, nitrogen adsorption–desorption isotherms (Figure 3) for AC–O series are type IV, AC–KB series are type II, and AC–N series are type II. The PSD plots determined for all obtained N-rich activated carbons are shown in Figure 4. These plots are obtained from the nonlocalized density functional theory (NLDFT) method in the SAIEUS program. The AC–O and AC–N series have similar pore widths from 1 to 2 nm. In turn, the AC–KB series have pore widths between 1–4 nm. The hysteresis loops have the same shape, and gamma irradiation and nitrogen enrichment have an effect on their shape. The morphology of the starting activated carbon material has a significant influence on the porosity results. The contribution of the mesopore volume V_me_ to the total pore volume V_t_ increased in the AC–O and AC–N series. Only for the AC–KB series, this value remains constant (99%). Figure 4 allows us to conclude that the proposed N-doping method does not considerably alter pore size distribution of treated carbon matrixes in comparison with raw carbon materials. It suggests that molecules of amine, presumably adsorbed in pores of treated carbons, are removed/desorbed in the last steps of the manufacturing procedure. The conclusion is supported by the data in Table 2 since S_BET_ and V_t_ values only underwent a minor decrease.

Raman spectroscopy was used to better characterize the obtained materials. Raman spectra and the intensity ratio of the D-band and G-band (I_D_/I_G_) is presented in Figure 5 and Table 3. Figure 5 shows a single sharp G peak at 1592 cm^−1^ and a characteristic band at 1345 cm^−1^ (D band). The ratio of intensities between the D and G bands (I_D_/I_G_) depends on the level of the disorder. The I_D_/I_G_ ratios for the AC–O series are between 0.98–1.01; the AC–KB series are between 1.11–1.24, and the AC–N series are between 1.06–1.11. The I_D_/I_G_ ratio did not change significantly. This means that the size of crystalline domains and the number of defects were comparable.

Next, XPS analysis was aimed at discovering the chemical environment of obtained samples. The XPS spectra of obtained samples are demonstrated in Figure 6, Figure 7 and Table 4. The analysis of C1s energy determined the chemical bonding of carbon atoms. The C1s spectra are composed of seven peaks corresponding to the C=C bond (sp^2^) peak at 284.4 eV [38], the C–C bond (sp^3^) peak at 284.9 eV [38,39], the C–O–C, C–OH or C–NH bond peak at 286.2 eV [38,39], the C=O, O–C–O or N–C–O bond peak at 287.5 eV [38,39] and the O–C=O bond peak at 288.6 eV [38]. Peaks 6 and 7 (290.1 eV and 293.4 eV, respectively) are related to the shake-up excitation [40]. The excitation of the shake-up type comes from the carbon sp^2^ and its aromatic forms. Moreover, it is an additional parameter confirming the presence of this type of bond [38,41]. The total amount of oxygen in samples is in the range from 0.7 to 5.0 at.%. The peak at 530.7 eV corresponds to the O*=C–O or O–C–O bond, whereas the second peak at 532.8 eV corresponds to O=C–O*, C–O–C or –OH [38,39,41].

The XPS spectra of representative samples (Figure 7) show two peaks at 399.1 and 401.3 eV, which are characteristic of nitrogen (N1s). Peaks at 399.1 eV correspond to C=N–R, CH_2_–NH_2_ or in the form of pyridine [39]. In turn, the peak at 401.3 eV corresponds to (C=O)–N–(C=O), -NH–(C=O)–O or the -NH- group attached to an aromatic ring [38].

The elemental content of nitrogen is in the range from 1.3 to 1.6 at.% for the AC–O series, 1.4 to 2.2 at.% for the AC–KB series and 1.1 to 1.4 at.% for the AC–N series, respectively. In turn, the total amount of oxygen in the samples is in the range of 0.7 to 5.0 at.%, respectively.

Moreover, for the AC–N series, samples show only one characteristic peak of nitrogen at 401.3 eV. The raw XPS spectra of the investigated samples are presented in Appendix A.

Structural parameters, i.e., the surface area, total pore volume, nitrogen content and the volume share ascribed to mesopores, assume that the obtained N-rich activated carbons are expected to be effective electrodes for electrochemistry tests. Therefore, the next step was electrochemical measurements.

### 3.2. Electrochemical Performance

The catalytic activity of the obtained carbon materials in the oxygen reduction reaction was tested. Cyclic voltammetry and linear voltammetry tests in oxygen-saturated and nitrogen-saturated 0.1 M KOH solutions were aimed at determining potential application in metal–air batteries or fuel cells. Table 5 presents the characteristic parameters for the samples obtained, such as the value of the cathode peak (E_p_), the onset potential (E_onset_), the half potential (E_1/2_), the diffusion-limiting current value and the number of electrons transferred in the oxygen reduction reaction. The presented CV curves in Figure 8 are correlated for a particular group of materials and compared with commercial carbon with a 20% addition of platinum. The cathode peak is clearly visible for the group of compounds based on active carbon (AC–O) and also (AC–KB) and is in the range of 0.69–0.70 V vs. RHE and 0.72–0.75 V vs. RHE, respectively. For the remaining groups of materials based on (AC–N), the peak is less visibly blurred compared with the previous group. The tested LSV measurements made it possible to determine the number of electrons carried on in the oxygen reduction reaction. The results shown in Figure 9 were compared for CV measurements to a commercial platinum-based carbon material at a rotating disc electrode (RDE) speed of 1600 rpm. The largest diffusion-limiting current was recorded for AC–O and AC–KB materials and in the range off 2.99–3.30 mA cm^−^^2^ and 2.80–2.99 mA cm^−^^2^, respectively. For the remaining AC–N materials, the ranges were 2.10–1.72 mA cm^−^^2^.

By analyzing the parameter diffusion-limiting current in Table 5, the relationship between radiation doses is visible. The highest values are observed after the first dose, while after the second dose of radiation, this parameter decreases, which could indicate the formation of functional groups that do not affect the oxygen reduction reaction. This also has an influence on the number of transferred electrons, which is an important parameter for determining the catalytic activity of the obtained materials, determined on the basis of the Koutecky–Levich equation. All obtained materials show a 2-electron oxygen reduction pathway. The materials AC–O-I and AC–O-II showed the highest number of transferred electrons and amounted to 3.09 and 2.77, respectively. For AC–KB and AC–N samples, the number of electrons range from 2.43 to 2.51 and from 1.95 to 2.21, respectively, and the comparison for other carbon materials and raw materials is shown in the graph of Figure 10.

Activated carbons are considered catalytically inactive materials in the oxygen reduction reaction; therefore, it is important to dope such materials with nitrogen in order to improve their catalytic activity. The number of functional groups is not important as much as the quality of the nitrogen groups produced. Even a small percentage of nitrogen can have a positive effect on ORR parameters, as can the functional groups located in the mesoporous carbon structure that create active sites for oxygen reduction, according to the reaction equations [42,43]:(3)$$O_{2\;\left(g\right)}+\;*\;\to O_{2}^{*}$$
(4)$$O_{2}^{*}+{\;H}_{2}O\;\left(I\right)+{\;e}^{-}\to {\mathrm{HOO}}^{*}+{\;\mathrm{OH}}^{-}$$
(5)$${\mathrm{HOO}}^{*}+{\;e}^{-}\to O^{*}+{\;\mathrm{OH}}^{-}$$
(6)$$O^{*}+{\;H}_{2}O\;\left(I\right)+{\;e}^{-}\to {\mathrm{HO}}^{*}+{\;\mathrm{OH}}^{-}$$
(7)$${\mathrm{HO}}^{*}+{\;e}^{-}\to {\mathrm{OH}}^{-}+\;*$$
(8)$$O_{2\;\left(g\right)}+\;*\;+\;*\;\to O^{*}+{\;O}^{*}$$

In the gamma irradiation method, ions, electrons and excited molecules are the primary, transient state products of the radiolysis of amines. Both ions and excited molecules give free radicals, products of the decomposition of exciting species or ion–molecule and neutralization reactions. The typical volatile products of pure amine radiolysis are hydrogen, methane and ammonia. The radiation yields of hydrogen and methane depend on the amine, with the highest value for methylamine and much less for secondary or tertiary amines. Free radicals can be formed at both nitrogen and carbon atoms and may react with amine or with one another. Irradiation of tert-butylamine gives products suggesting the formation of both ButNH* and NH_2_CMe_2_CH_2_* radicals [44,45], which can dimerise or couple.

## 4. Conclusions

Some intensively N-doped mesoporous activated carbons were successfully prepared using gamma irradiation at a radiation dose lower than 6 kGy from commercial precursors. The proposed gamma irradiation method is universal for different materials and did not destroy the original morphology of the used activated carbons. In the current study, the authors proved the applicability of activated carbons as an extension to previously verified applicability to graphene/graphite. SEM/EDX mapping, nitrogen adsorption data, elemental analysis and XPS confirm that N-atoms were in the structure of the investigated activated carbons. Nitrogen enrichment proceeded in a controlled manner i.e., the N-content increased upon the irradiation dose.

Generally, nitrogen can positively influence the electrochemical properties of materials. The electrochemical results show that not all used and modified activated carbons are essential from an electrochemical point of view. Based on our results, it can be concluded that only the AC–O series may potentially be a promising electrode material. However, these samples have the lowest nitrogen content. Hence, high N-content or a large surface area does not always have a positive effect on the electrochemical properties of the final material.

We have proved that high nitrogen content is not always enough to improve a given material. In our previous paper [16], we demonstrated that the N-insertion issue is a very complex one. We found that structural factors may play as crucial a role as N-doping itself in the improvement of electrochemical performance. We intend to resolve the problem and investigate it intensively. The current paper is the next argument in favor of the statement that N-doping is only one of the key factors governing the electrochemical performance. Our work also points out which of the commercially available activated carbons is worth N-doping aiming at better behavior in ORR (AC–O).

The current research again points out the complexity of N-doping of diversified carbon matrixes and the fact that the quantity of N is not the sole factor governing the electrochemical performance of such materials in ORR.

The aspect of using the proposed self-designed gamma irradiation method for introducing other heteroatoms to different carbon materials will be continued in further studies.

## Figures and Tables

**Figure 1 nanomaterials-12-03156-f001:**
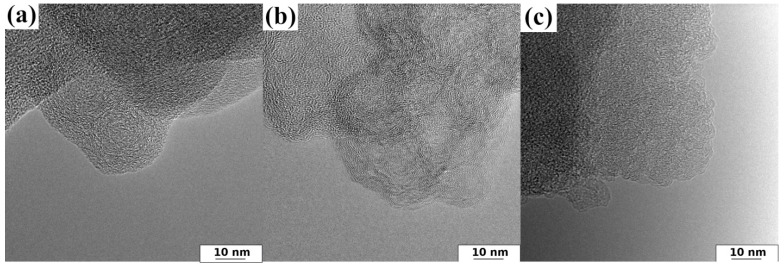
HRTEM images of representative samples for all investigated series of the obtained materials: (**a**) AC–O; (**b**) AC–KB; (**c**) AC–N.

**Figure 2 nanomaterials-12-03156-f002:**
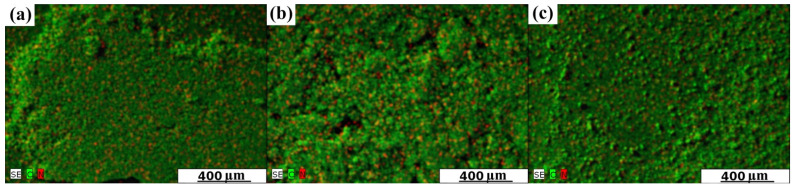
SEM mapping images of all investigated series of the obtained samples: (**a**) AC–O-II; (**b**) AC–KB-II; and (**c**) AC–N-II.

**Figure 3 nanomaterials-12-03156-f003:**
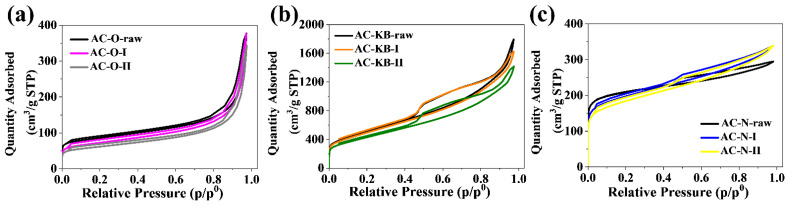
Nitrogen adsorption–desorption isotherms of all investigated series of the obtained samples: (**a**) AC–O; (**b**) AC–KB; and (**c**) AC–N.

**Figure 4 nanomaterials-12-03156-f004:**
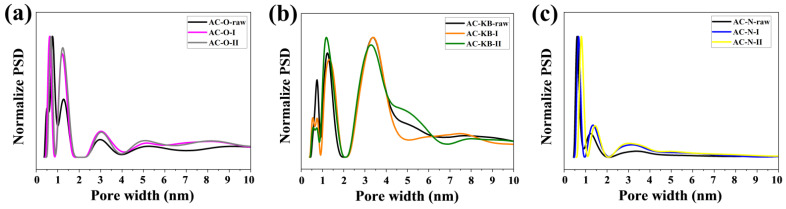
Pore size distribution obtained from the adsorption branches of N_2_ used the nonlocalized density functional theory (NDLFT) method of all investigated series of the obtained samples: (**a**) AC–O; (**b**) AC–KB; and (**c**) AC–N.

**Figure 5 nanomaterials-12-03156-f005:**
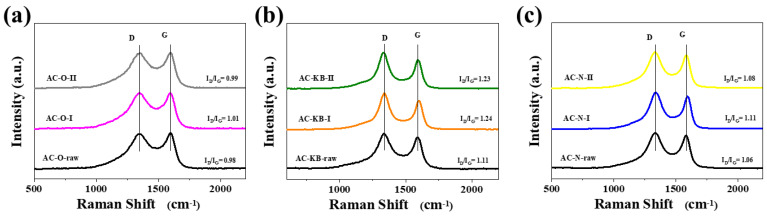
Raman spectra of all investigated series of the obtained samples: (**a**) AC–O; (**b**) AC–KB; and (**c**) AC–N.

**Figure 6 nanomaterials-12-03156-f006:**
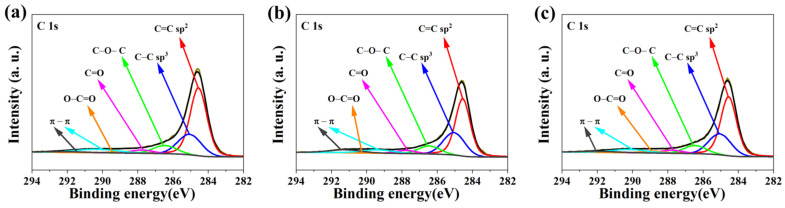
XPS spectra for C1s of representative samples of the obtained N-rich activated carbons: (**a**) AC–O-II; (**b**) AC–KB-II; and (**c**) AC–N-II.

**Figure 7 nanomaterials-12-03156-f007:**
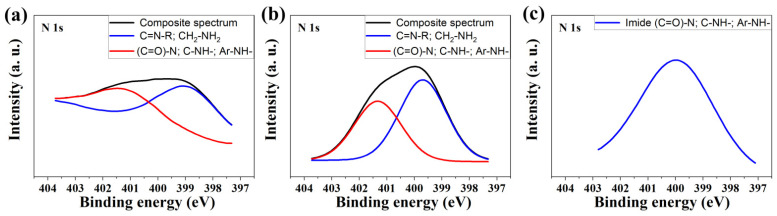
XPS spectra for N1s of representative samples of the obtained N-rich activated carbons: (**a**) AC–O-II; (**b**) AC–KB-II; and (**c**) AC–N-II.

**Figure 8 nanomaterials-12-03156-f008:**
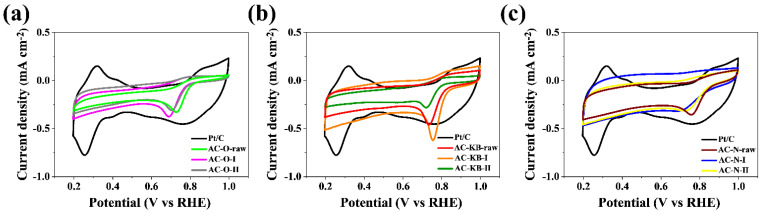
CV curves of the (**a**) AC–O; (**b**) AC–KB; (**c**) AC–N compared with Pt/C catalysts in an O_2_-saturated 0.1 M KOH solution.

**Figure 9 nanomaterials-12-03156-f009:**
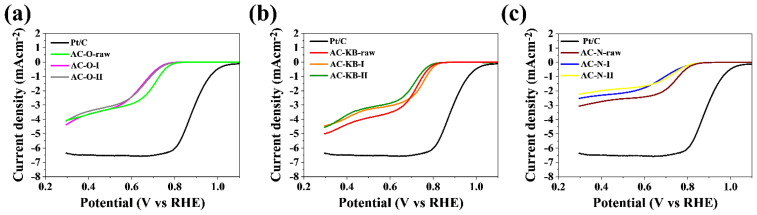
LSV curves of the (**a**) AC–O; (**b**) AC–KB; (**c**) AC–N compared with Pt/C catalysts, at a scan rate of 5 mV s^−1^ and a rotation rate of 1600 rpm in O_2_-saturated alkaline media.

**Figure 10 nanomaterials-12-03156-f010:**
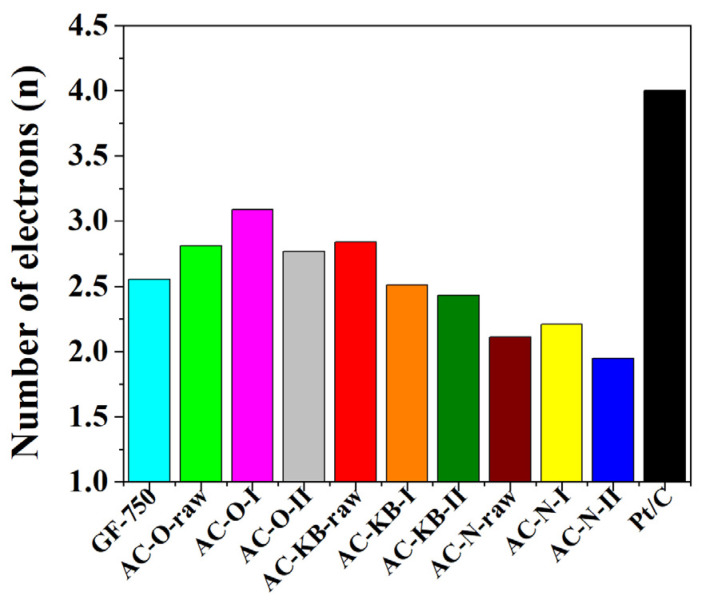
The number of transfer electrons in ORR for all samples compared with commercial Pt/C and commercial graphene nanoplatelets GF-750.

**Table 1 nanomaterials-12-03156-t001:** An overview of currently practiced methods for nitrogen insertion into the relatively chemically inert graphene matrix [17].

Method	Description	Drawbacks	Example Study
CVD	High-temperature furnace up to 1000 °C, vacuum 1 Torr; catalyst; NH_3_ as a nitrogen source; He as a shielding gas	Complex instrumentation; exceptionally low yield	[19,20]
Arc Discharge	Electric arc discharge conditions; pyridine and a NH_3_ as nitrogen source	Complex instrumentation; difficult to control; exceptionally low yield	[21]
Pyrolysis	High-temperature pyrolysis of a solid mixture of GO-urea	Limited yield; long time high-temperature synthesis; application of GO instead of pure graphene	[22]
Heat treating	Heating to 800–1000 °C of a solid mixture of the GO-nitrogen source; a neutral atmosphere; melamine as a potential nitrogen source	Limited yield; long time high-temperature synthesis; the application of GO instead of pure graphene	[23]
Solvothermal	200–300 °C; 4–5 h duration; dimethylformamide as a solvent and nitrogen source	Yield limited by the experimental vessel volume; use environmentally and health unfriendly reagents	[24]
Gas Annealing	A high temperature of 500–1000 °C during electrical annealing of GO in the nitrogen atmosphere; ammonia gas (NH_3_) as a nitrogen source	GO applied instead of pure graphene/graphite; low yield; a high-temperature method	[25,26]
N_2_ Plasma Treatment	The nitrogen content is controlled by the plasma strength and exposure time; example plasma generator parameters 40–200 W; 900 V DC bias; high vacuum 200 mTorr; 20–80 min treatment; Graphene or GO as a key precursor; N_2_ and NH_3_ as nitrogen source	Sophisticated instrumentation and challenging synthesis conditions; low yield	[27]
Dry Ball Milling	A mechano-chemical process; room temperature direct grinding of dry powdered graphite in the N_2_ or NH_3_ atmosphere; a nitrogen content controllable by changing milling parameters	The unwanted insertion of impurities from the grinding setup which must be removed by additional treatment; a laboratory-scale process	[28]
Nanoscale High Energy Wet Ball Milling	A mechano-chemical process; room to 80 °C wet milling; gas, liquid, and solid nitrogen carriers acceptable; GO advised as a carbon precursor	A complex manufacturing pathway including frequent rising, a laboratory-scale process	[29]
MSCC-ET	The molten salt carbon capture and electrochemical transformation method can create N-doped carbon on a scale of hundreds of kilograms; yield highly ORR-active catalysts	high-temperature; eventual impurities	[30,31,32]

**Table 2 nanomaterials-12-03156-t002:** The chemical composition and structural parameters of the obtained N-doped activated carbons and raw materials.

Sample	Elemental Content (wt.%)				S_BET_	V_t_	V_mi_	V_me_	V_me_/V_t_
	N	C	H	Residue	(m^2^/g)	(cm^3^/g)	(cm^3^/g)	(cm^3^/g)	(%)
AC–O-raw	0.4	96.3	0.5	2.8	291	0.584	0.065	0.519	89
AC–O-I	0.9	95.2	1.0	2.9	248	0.583	0.043	0.540	93
AC–O-II	1.3	95.7	0.8	2.2	223	0.531	0.036	0.495	93
AC–KB-raw	0.2	97.3	0.4	2.1	1837	2.773	0.016	2.754	99
AC–KB-I	2.9	94.2	1.0	1.9	1787	2.508	0.014	2.494	99
AC–KB-II	3.2	94.7	1.0	1.1	1556	2.195	0.026	2.169	99
AC–N-raw	0.8	87.0	1.0	11.2	705	0.456	0.256	0.200	44
AC–N-I	1.6	85.2	1.1	12.1	661	0.524	0.176	0.348	66
AC–N-II	1.8	85.7	1.0	11.5	625	0.525	0.153	0.372	71

**Table 3 nanomaterials-12-03156-t003:** D and G band data of Raman spectra.

Sample	I_D_	cm^−1^	I_G_	cm^−1^	I_D_/I_G_
AC–O-raw	772	1345	739	1597	0.98
AC–O-I	532	1347	529	1592	1.01
AC–O-II	605	1349	611	1596	0.99
AC–KB-raw	750	1336	677	1591	1.11
AC–KB-I	1426	1338	1149	1599	1.24
AC–KB-II	926	1331	753	1592	1.23
AC–N-raw	3136	1335	2953	1582	1.06
AC–N-I	2908	1338	2628	1593	1.11
AC–N-II	8087	1330	7472	1584	1.08

**Table 4 nanomaterials-12-03156-t004:** The elemental composition of the obtained N-doped activated carbons from XPS spectra.

Element	C								O			N		
Binding energy (eV)	284.4	284.9	286.2	287.5	288.6	290.1	293.4		530.7	532.8		399.1	401.3	
Sample	Content (at.%)							% of total	Content (at.%)		% of total	Content (at.%)		% of total
AC–O-raw	54.5	22.8	8.0	3.3	2.6	4.8	2.6	98.6	0.7	0.3	1.0	0.2	0.2	0.4
AC–O-I	52.0	23.2	8.6	3.2	1.4	7.0	2.1	97.5	0.2	1.0	1.2	0.8	0.5	1.3
AC–O-II	55.3	21.0	8.2	3.2	1.7	6.2	1.3	96.9	0.4	1.1	1.5	0.9	0.7	1.6
AC–KB-raw	50.6	23.8	8.9	3.9	1.7	7.3	3.1	99.3	0.3	0.4	0.7	0.0	0.0	0.0
AC–KB-I	51.3	23.0	8.3	3.5	1.0	8.2	2.6	97.9	0.4	0.3	0.7	0.9	0.5	1.4
AC–KB-II	50.3	23.4	9.0	3.7	0.0	5.2	5.2	96.8	0.7	0.3	1.0	1.4	0.8	2.2
AC–N-raw	50.7	22.4	8.8	3.3	2.1	5.8	1.9	95.0	2.5	2.5	5.0	0.0	0.0	0.0
AC–N-I	42.9	25.4	9.6	4.2	2.8	8.0	2.0	94.9	2.2	1.8	4.0	0.0	1.1	1.1
AC–N-II	50.2	22.3	8.7	3.2	2.2	6.2	1.9	94.7	2.3	1.6	3.9	0.0	1.4	1.4

**Table 5 nanomaterials-12-03156-t005:** ORR performance parameters of the obtained N-doped activated carbons and commercial Pt/C catalysts tested in alkaline media.

Catalyst	E_p_ (V vs. RHE)	E_onset_ (V vs. RHE)	E_1/2_ (V vs. RHE)	Diffusion-Limiting Current (mA cm^−2^)	n (0.5 V)
Pt/C	0.76	0.98	0.88	6.37	4.00
AC–O-raw	0.73	0.77	0.71	2.93	2.81
AC–O-I	0.69	0.74	0.66	3.30	3.09
AC–O-II	0.70	0.75	0.66	2.99	2.77
AC–KB-raw	0.73	0.79	0.72	3.44	2.84
AC–KB-I	0.75	0.81	0.75	2.99	2.51
AC–KB-II	0.72	0.76	0.71	2.80	2.43
AC–N-raw	0.75	0.81	0.75	2.36	2.11
AC–N-I	0.68	0.80	0.69	2.10	2.21
AC–N-II	0.72	0.82	0.73	1.72	1.95

## Data Availability

The data presented in this study are available on request from the corresponding authors.

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
