# Peer review of "Obtaining N-Enriched Mesoporous Carbon-Based by Means of Gamma Radiation"

_nanomaterials, 2022, doi:10.3390/nano12183156_

Round 1

Reviewer 1 Report

In this paper, the authors purchased three commercially available activated carbons to obtain N-doped carbon materials by means of gamma radiation. To prove that the N element has been doped successfully, the authors performed some conventional characterization. Finally, N-doped activated carbon was used for electrochemical studies. After reading this paper, I cannot suggest the publication of this paper for the following reasons.

1.     The authors hoped to achieve enhanced electrochemical properties of carbon materials by doping commercial activated carbons with N elements, but only two of them showed only a slight enhancement, and the remaining one even exhibited a continuous downward trend. Such results are unconvincing.

2.     None of the three commercial activated carbons showed the highest values of N content after radiation. Is it possible that the slight fluctuations in the electrochemical test results that follow are just systematic errors? Is it possible to continue to increase the irradiation and radiation dose to find the peak N content, as well as to supplement the ORR tests?

3.     All characterization results of the carbon material before and after the γ-radiation treatment was similar. Curiously, the authors did not compare the HRTEM images before and after the treatment, hoping to add this part to demonstrate the surface morphological changes of the activated carbon.

4.     “N2” appears in many places in the article and should be corrected to “N2”.

5.     In the introduction section, there are several places where the source of the citation is not marked.

Author Response

On behalf of all authors, I would like to express our gratitude for a careful reading of our manuscript and subsequent constructive remarks. We intend to improve our manuscript in accordance with these comments. Please see our point-to-point responses to the reviewers’ comments, marked in red below. The corresponding changes to the comments are highlighted with a yellow background in the revision.

Sincerely yours,

PhD Piotr Kamedulski, corresponding author

REVIEWER 1

In this paper, the authors purchased three commercially available activated carbons to obtain N-doped carbon materials by means of gamma radiation. To prove that the N element has been doped successfully, the authors performed some conventional characterization. Finally, N-doped activated carbon was used for electrochemical studies. After reading this paper, I cannot suggest the publication of this paper for the following reasons.

  1. The authors hoped to achieve enhanced electrochemical properties of carbon materials by doping commercial activated carbons with N elements, but only two of them showed only a slight enhancement, and the remaining one even exhibited a continuous downward trend. Such results are unconvincing.

Response: We thank the Reviewer for this remark. We have proved that high nitrogen content is not always enough to improve a given material. In our paper Kamedulski et al. “Materials 14(2021) Article number 2448; “The importance of structural factors for the electrochemical performance of graphene/carbon nanotube/melamine powders towards the catalytic activity of oxygen reduction reaction” we have demonstrated that the N-insertion issue is a very complex one. Finally, we have concluded that structural factors may play as crucial role as N-doping itself in the improvement of electrochemical performance. In the revised version we have added a comment on the diversified influence on N-doping. The porous structure is also important, especially the presence of mesopores and the increased surface area. We intend to resolve the problem, and the current paper is just a next argument in favor of the statement that N-doping is only one of the key factors governing the electrochemical performance. Our work is also pointing out, which of commercially available activated carbons is worth of N-doping aiming at the better behavior in ORR.

  1. None of the three commercial activated carbons showed the highest values of N content after radiation. Is it possible that the slight fluctuations in the electrochemical test results that follow are just systematic errors? Is it possible to continue to increase the irradiation and radiation dose to find the peak N content, as well as to supplement the ORR tests?

Response: We thank the Reviewer for this remark. However, we can not agree that the gamma irradiation did not increase the nitrogen content. All three irradiated carbons (AC-O, AC-KB, AC-N denoted by “-I” and “-II” suffix – Table 4) exhibited multiplicated nitrogen content after radiation (see Table 4). For example, specimen A-CN (material before radiation), N content was 0.0% at., but after the first irradiation it increased to 1.1% at. (sample AC-N-I), and a further increase was observed to 1.4% at. (sample AC-N-II). The same qualitative effect was observed for two other carbon i.e. AC-O and AC-KB. Thus, the performed gamma radiation based N-doping is effective, however not spectacular and the final N-content was not as high as in case of other doping methods (Kucinska et al. Carbon 50(2012)3098 “A microporous and high surface area active carbon obtained by the heat-treatment of chitosan”). These very effective/spectacular methods in most rely on a complete synthesis of N-rich activated carbons i.e. the high N-content was achieved by carbonization of N-rich biopolymers (chitin, chitosan, amino acids, etc.), which is very far form our concept. We intended to modify (N-doping) of already manufactured typical ACs. Such an approach may be N-doped by ammonia treatment (Horikawa et al., Carbon 50(2012)1833-1842 “Preparation of nitrogen-doped porous carbon by ammonia gas treatment and the effects of N-doping on water adsorption”), which suffers from two basic shortcomings: high mass loss and the emission of toxic HCN. We are just searching for (and testing the limits) an alternative method i.e. the gamma radiation based.

We have added some comments to the manuscript.       

  1. All characterization results of the carbon material before and after the γ-radiation treatment was similar. Curiously, the authors did not compare the HRTEM images before and after the treatment, hoping to add this part to demonstrate the surface morphological changes of the activated carbon.

Response: We thank the Reviewer for this remark. We add the HRTEM images in different places and under different magnifications for the sample before the treatment in Supplementary Information. HRTEM belongs to typical carbon characterization methods. However, such images are not very informative in case of heteroatom insertion to the carbon matrix, which is an atomic scale effect. Even very high resolution of some HRTEM instruments does not let to see significant changes in the carbon matric structure and morphology.   

  1. “N2” appears in many places in the article and should be corrected to “N2”.

Response: We thank the Reviewer for this remark. We have corrected this error.

  1. In the introduction section, there are several places where the source of the citation is not marked.

Response: We thank the Reviewer for this remark. We have corrected this error.

Reviewer 2 Report

This manuscript report a method for preparation of N-doped carbons by gamma irradiation. The enrichment of N species has been confirmed by XPS. The impact on the electrocatalytic activity is not so high, but the series of data would deserved to be shared in the community. 

This catalyst is rather active for electrochemical reduction of O2 to for H2O2, which is also a very important electrochemical reduction. Citing the following papers could strengthen the introduction part.

a. 10.1021/acs.jpcc.8b12464

b. 10.1016/0013-4686(90)87004-L

c.  10.1021/ja300038p

Author Response

On behalf of all authors, I would like to express our gratitude for a careful reading of our manuscript and subsequent constructive remarks. We intend to improve our manuscript in accordance with these comments. Please see our point-to-point responses to the reviewers’ comments, marked in red below. The corresponding changes to the comments are highlighted with a yellow background in the revision.

Sincerely yours,

PhD Piotr Kamedulski, corresponding author

REVIEWER 2

This manuscript report a method for preparation of N-doped carbons by gamma irradiation. The enrichment of N species has been confirmed by XPS. The impact on the electrocatalytic activity is not so high, but the series of data would deserved to be shared in the community. 

This catalyst is rather active for electrochemical reduction of O2 to for H2O2, which is also a very important electrochemical reduction. Citing the following papers could strengthen the introduction part.

  1. 10.1021/acs.jpcc.8b12464
  2. 10.1016/0013-4686(90)87004-L
  3. 10.1021/ja300038p

Response: We thank the Reviewer for this remark. The introduction section has been enhanced. We have added new literature references in the Introduction (red color in manuscript).

Reviewer 3 Report

This manuscript deals with a novel way of creating doped carbon materials: using gamma-radiation to include nitrogen from surrounding tert-butylamine. The method is quite novel and the authors have presented a physico-chemical analysis of materials doped using two radiation doses, which allows to make some conclusions on the effect of the radiation. However, the data is presented incompletely in places (specifically the XPS spectra) and could certainly be improved. The main use case the authors have presented for the materials, their ORR activity, is also rather low when compared to the literature. However, the method remains interesting and the manuscript would be of interest to the readers of Nanomaterials if improvements are made. Specific issues:

1)      The authors mainly focus on the scale-up advantage of their method. What would the pricing be for the obtained carbon material on the kilogram-scale?

2)      The authors describe methods such as CVD and arc discharge to require “complex instrumentation”. What about the instrumentation required for using gamma radiation for doping carbon, especially considering building a safe containment unit? How does the capital investment for the instrumentation compare to, for example, arc discharge and CVD methods?

3)      The authors have described methods such as solvothermal N-doping using “environmentally and health unfriendly reagents”. How about the environmental and health effects of tert-butylamine, the main reagent for the authors?

4)      How much carbon can be produced with the author’s method and has this kind of doping been used on the large scale anywhere else?

5)      One method the authors have not discussed that can also yield doped carbon is the molten salt carbon capture and electrochemical transformation (MSCC-ET) method. This method can create nitrogen doped carbon (see M. Johnson, J. Ren, M. Lefler, G. Licht, J. Vicini, S. Licht, Data Br. 14 (2017) 592–606. doi:10.1016/J.DIB.2017.08.013) on a scale of hundreds of kilograms (S. Licht, X. Liu, G. Licht, X. Wang, A. Swesi, Y. Chan, , Mater. Today Sustain. 6 (2019) 100023. doi:10.1016/j.mtsust.2019.100023) and has already also been demonstrated to yield highly ORR-active catalysts (see S. Ratso, P.R. Walke, V. Mikli, J. Ločs, K. Šmits, V. Vītola, A. Šutka, I. Kruusenberg, Green Chem. 23 (2021) 4435–4445. doi:10.1039/d1gc00659b and A. Remmel, S. Ratso, G. Divitini, M. Danilson, V. Mikli, M. Uibu, J. Aruväli, I. Kruusenberg, Nickel and Nitrogen-Doped Bifunctional ORR and HER Electrocatalysts Derived from CO 2, ACS Sustain. Chem. Eng. 10 (2021) 134–145. doi:10.1021/acssuschemeng.1c05250). Discussion on this method should be added Table 1 and the relevant articles referenced.

6)      The PSD graphs should be superset on top of each other with a common y-axis in Figure 4. Currently it is very difficult to see the minor differences in the PSD of the materials due to the layout.

7)      The XPS data should also include the raw spectra, as currently it is impossible to see the signal-to-noise ratio.

8)      How reproducible is the XPS deconvolution for the samples with <1.5 at.% content? How many times were these experiments repeated?

9)      The authors have written that “activated carbons are considered catalytically inactive materials in the oxygen reduction reaction; therefore, it is important to dope such materials with nitrogen in order to improve their catalytic activity”. However, most of the “raw” carbons that they use, have either a more positive onset potential (in the case of AC-O and AC-N), or even a higher n (in the case of AC-KB) than their doped variants. Clearly, in this case, nitrogen doping even inhibits the materials’ ORR activity! How can this be explained?

10)   Topical references on the gamma-radiation induced doping of carbon such as T. Thurakitseree, C. Kramberger, N. Chanlek, R. Supruangnet, A. Wisitsorraat, H. Nakajima, S. Maruyama, Diam. Relat. Mater. 101 (2020) 107569. doi:10.1016/J.DIAMOND.2019.107569; T. Thurakitseree, C. Kramberger, N. Chanlek, H. Nakajima, Possibility of doping nitrogen into single-walled carbon nanotubes by γ-irradiated N2 molecules, Radiat. Phys. Chem. 186 (2021) 109524. doi:10.1016/J.RADPHYSCHEM.2021.109524; should be added.

Author Response

On behalf of all authors, I would like to express our gratitude for a careful reading of our manuscript and subsequent constructive remarks. We intend to improve our manuscript in accordance with these comments. Please see our point-to-point responses to the reviewers’ comments, marked in red below. The corresponding changes to the comments are highlighted with a yellow background in the revision.

Sincerely yours,

PhD Piotr Kamedulski, corresponding author

REVIEWER 3

This manuscript deals with a novel way of creating doped carbon materials: using gamma-radiation to include nitrogen from surrounding tert-butylamine. The method is quite novel and the authors have presented a physico-chemical analysis of materials doped using two radiation doses, which allows to make some conclusions on the effect of the radiation. However, the data is presented incompletely in places (specifically the XPS spectra) and could certainly be improved. The main use case the authors have presented for the materials, their ORR activity, is also rather low when compared to the literature. However, the method remains interesting and the manuscript would be of interest to the readers of Nanomaterials if improvements are made. Specific issues:

1)      The authors mainly focus on the scale-up advantage of their method. What would the pricing be for the obtained carbon material on the kilogram-scale?

Response: We thank the Reviewer for this remark. Gamma radiation bomb, in general and case of our research, is a specific instrument, which operation does not require much power consumption. The gamma source emits the radiation constantly, and only a little diminishing of its intensity may be observed. Our bomb is in operation for ca. 40 years using the same original gamma source. Thus, for our experiments we used the gamma bomb and no cost was attributed to the experiment. The real cost was paid ca. 40 years ago when the whole installation (the gamma bomb) was bought. Therefore, the whole cost of the experiment is the cost of original activated carbons and N-doping reactants; it is neglectable. The method is cost-effective provided one has an access to analogous already working gamma devices.      

2)      The authors describe methods such as CVD and arc discharge to require “complex instrumentation”. What about the instrumentation required for using gamma radiation for doping carbon, especially considering building a safe containment unit? How does the capital investment for the instrumentation compare to, for example, arc discharge and CVD methods?

Response: We thank the Reviewer for this remark. Please, see the answer for question 1).

3)      The authors have described methods such as solvothermal N-doping using “environmentally and health unfriendly reagents”. How about the environmental and health effects of tert-butylamine, the main reagent for the authors?

Response: We thank the Reviewer for this remark. In this study we used the butylamine, which is risky for human health and environment. It is possible to use other sources for N for the carbon matrix doping, which are less dangerous as water solution of ammonia (doping in liquid phase). We have revised the text to avoid the misunderstanding.

4) How much carbon can be produced with the author’s method and has this kind of doping been used on the large scale anywhere else?

Response: We thank the Reviewer for this remark. This gamma bomb we have used has a rather small working chamber, which volume is ca. 1 liter. Thus, any up-scaling of the process with this device is limited. However, there are some industrial, high-capacity devices as well (like gamma sterilization autoclaves).   

5) One method the authors have not discussed that can also yield doped carbon is the molten salt carbon capture and electrochemical transformation (MSCC-ET) method. This method can create nitrogen doped carbon (see M. Johnson, J. Ren, M. Lefler, G. Licht, J. Vicini, S. Licht, Data Br. 14 (2017) 592–606. doi:10.1016/J.DIB.2017.08.013) on a scale of hundreds of kilograms (S. Licht, X. Liu, G. Licht, X. Wang, A. Swesi, Y. Chan, , Mater. Today Sustain. 6 (2019) 100023. doi:10.1016/j.mtsust.2019.100023) and has already also been demonstrated to yield highly ORR-active catalysts (see S. Ratso, P.R. Walke, V. Mikli, J. Ločs, K. Šmits, V. Vītola, A. Šutka, I. Kruusenberg, Green Chem. 23 (2021) 4435–4445. doi:10.1039/d1gc00659b and A. Remmel, S. Ratso, G. Divitini, M. Danilson, V. Mikli, M. Uibu, J. Aruväli, I. Kruusenberg, Nickel and Nitrogen-Doped Bifunctional ORR and HER Electrocatalysts Derived from CO 2, ACS Sustain. Chem. Eng. 10 (2021) 134–145. doi:10.1021/acssuschemeng.1c05250). Discussion on this method should be added Table 1 and the relevant articles referenced.

Response: We thank the Reviewer for this remark. The introduction section has been enhanced. We have added new literature references in the Introduction (red color in manuscript).

6) The PSD graphs should be superset on top of each other with a common y-axis in Figure 4. Currently it is very difficult to see the minor differences in the PSD of the materials due to the layout.

Response: We thank the Reviewer for this remark. We change this image.

Figure 4. Pore size distribution obtained from the adsorption branches of N2 used the nonlocalized density functional theory (NDLFT) method of all investigated series of the obtained samples: a) AC-O, b) AC-KB, and c) AC-N

7) The XPS data should also include the raw spectra, as currently it is impossible to see the signal-to-noise ratio.

Response: We thank the Reviewer for this remark. We made the change as suggested by the reviewer. We add raw spectra in Supplementary Information.

8) How reproducible is the XPS deconvolution for the samples with <1.5 at.% content? How many times were these experiments repeated?

Response: We thank the Reviewer for this remark. XPS is strictly a surface method. Standard deviation of the XPS method is 0.5 at.%. Experiment was repeated two times.

9) The authors have written that “activated carbons are considered catalytically inactive materials in the oxygen reduction reaction; therefore, it is important to dope such materials with nitrogen in order to improve their catalytic activity”. However, most of the “raw” carbons that they use, have either a more positive onset potential (in the case of AC-O and AC-N), or even a higher n (in the case of AC-KB) than their doped variants. Clearly, in this case, nitrogen doping even inhibits the materials’ ORR activity! How can this be explained?

Response: We thank the Reviewer for this remark. We have proved that high nitrogen content is not always enough to improve a given material. The porous structure is also important, especially the presence of mesopores and the increased surface area. In the case of electrochemical measurements, the largest surface area does not guarantee the achievement of a good results too what we have proved in our previous work [P. Kamedulski et al., Materials 14(9), 2448, 2021]. Our work will be continue to significantly increase the surface of the material.

10) Topical references on the gamma-radiation induced doping of carbon such as T. Thurakitseree, C. Kramberger, N. Chanlek, R. Supruangnet, A. Wisitsorraat, H. Nakajima, S. Maruyama, Diam. Relat. Mater. 101 (2020) 107569. doi:10.1016/J.DIAMOND.2019.107569; T. Thurakitseree, C. Kramberger, N. Chanlek, H. Nakajima, Possibility of doping nitrogen into single-walled carbon nanotubes by γ-irradiated N2 molecules, Radiat. Phys. Chem. 186 (2021) 109524. doi:10.1016/J.RADPHYSCHEM.2021.109524; should be added.

Response: We thank the Reviewer for this remark. The introduction section has been enhanced. We have added new literature references in the Introduction (red color in manuscript).

Round 2

Reviewer 1 Report

All suggestiongs have been revised. The manuscript can be accepted for publication.

Reviewer 3 Report

All of the comments and suggestions have been sufficiently considered and the manuscript has been improved. I can now support publication in Nanomaterials.